# Single Cell Genetic Profiling of Tumors of Breast Cancer Patients Aged 50 Years and Older Reveals Enormous Intratumor Heterogeneity Independent of Individual Prognosis

**DOI:** 10.3390/cancers13133366

**Published:** 2021-07-05

**Authors:** Anna-Sophie Liegmann, Kerstin Heselmeyer-Haddad, Annette Lischka, Daniela Hirsch, Wei-Dong Chen, Irianna Torres, Timo Gemoll, Achim Rody, Christoph Thorns, Edward Michael Gertz, Hendrik Alkemade, Yue Hu, Jens K. Habermann, Thomas Ried

**Affiliations:** 1Section of Translational Surgical Oncology and Biobanking, Department of Surgery, University of Lübeck and University Medical Center Schleswig-Holstein, 23562 Lübeck, Germany; Anna-Sophie.Liegmann@uksh.de (A.-S.L.); alischka@ukaachen.de (A.L.); timo.gemoll@uni-luebeck.de (T.G.); halkemade@ukaachen.de (H.A.); 2Genetics Branch, Center for Cancer Research, National Cancer Institute, National Institutes of Health, Bethesda, MD 20892, USA; heselmek@mail.nih.gov (K.H.-H.); daniela.hirsch@nih.gov (D.H.); chenw4@mail.nih.gov (W.-D.C.); irianna_torres@urmc.rochester.edu (I.T.); yue.hu@nih.gov (Y.H.); 3Institute of Pathology, University Medical Center Mannheim, Medical Faculty Mannheim, Heidelberg University, 68167 Mannheim, Germany; 4Department of Gynecology and Obstetrics, Campus Lübeck, University Hospital of Schleswig-Holstein, 23562 Lübeck, Germany; Achim.Rody@uksh.de; 5Institute of Pathology, Marienkrankenhaus Hamburg, 22087 Hamburg, Germany; thorns.patho@marienkrankenhaus.org; 6Institute of Pathology, University of Lübeck and University Medical Center Schleswig-Holstein, 23562 Lübeck, Germany; 7Cancer Data Science Laboratory, Center for Cancer Research, National Cancer Institute, National Institutes of Health, Bethesda, MD 20892, USA; michael.gertz@nih.gov; 8Department of Oncology-Pathology, Cancer Center Karolinska, Karolinska Institute, 171 77 Stockholm, Sweden

**Keywords:** breast cancer, genomic instability, ploidy, copy number alterations (CNAs), inter- and intratumor heterogeneity (ITH), prognosis, next generation sequencing (NGS), multiplex fluorescence in situ hybridization (miFISH)

## Abstract

**Simple Summary:**

The majority (69.7%) of women diagnosed with breast cancer are above the age of 55. The population of older breast cancer patients is growing. Nevertheless, older patients are underrepresented in cancer research. Therefore, our study set the focus on breast cancer patients aged 50 years and older with a median age of 67 years aiming to understand the influence of aneuploidy, genomic instability and inter- and intratumor heterogeneity on disease outcome, being a major obstacle for precise prognostication and successful treatment. We analyzed chromosomal copy number changes, ploidy and specific gene mutations and found an enormous degree of genomic instability and intratumor heterogeneity in our cohort. However, neither the ploidy, the degree of intratumor heterogeneity nor the presence of specific gene mutations was correlated with prognosis. Our findings provide a precise description of the degree of intratumor heterogeneity, genomic instability, and gene mutations in breast cancer patients aged 50 years and older, revealing significant differences between diploid and aneuploid tumors regarding copy number alterations and the extent of genomic instability.

**Abstract:**

Purpose: Older breast cancer patients are underrepresented in cancer research even though the majority (81.4%) of women dying of breast cancer are 55 years and older. Here we study a common phenomenon observed in breast cancer which is a large inter- and intratumor heterogeneity; this poses a tremendous clinical challenge, for example with respect to treatment stratification. To further elucidate genomic instability and tumor heterogeneity in older patients, we analyzed the genetic aberration profiles of 39 breast cancer patients aged 50 years and older (median 67 years) with either short (median 2.4 years) or long survival (median 19 years). The analysis was based on copy number enumeration of eight breast cancer-associated genes using multiplex interphase fluorescence in situ hybridization (miFISH) of single cells, and by targeted next-generation sequencing of 563 cancer-related genes. Results: We detected enormous inter- and intratumor heterogeneity, yet maintenance of common cancer gene mutations and breast cancer specific chromosomal gains and losses. The gain of *COX2* was most common (72%), followed by *MYC* (69%); losses were most prevalent for *CDH1* (74%) and *TP53* (69%). The degree of intratumor heterogeneity did not correlate with disease outcome. Comparing the miFISH results of diploid with aneuploid tumor samples significant differences were found: aneuploid tumors showed significantly higher average signal numbers, copy number alterations (CNAs) and instability indices. Mutations in *PIKC3A* were mostly restricted to luminal A tumors. Furthermore, a significant co-occurrence of CNAs of *DBC2/MYC*, *HER2/DBC2* and *HER2/TP53* and mutual exclusivity of CNAs of *HER2* and *PIK3CA* mutations and CNAs of *CCND1* and *PIK3CA* mutations were revealed. Conclusion: Our results provide a comprehensive picture of genome instability profiles with a large variety of inter- and intratumor heterogeneity in breast cancer patients aged 50 years and older. In most cases, the distribution of chromosomal aneuploidies was consistent with previous results; however, striking exceptions, such as tumors driven by exclusive loss of chromosomes, were identified.

## 1. Introduction

Next to non-melanoma skin cancer, breast cancer is the most common cancer in women with an estimated 276,480 new cases diagnosed in the United States in 2020 [1].

One in eight women in the United States will develop breast cancer over an 80-year lifespan. The risk of developing breast cancer depends on a combination of factors, with advancing age playing a major role: the incidence of breast cancer rises dramatically with age [1]. Breast cancer is most-frequently diagnosed among women aged 55–74 with an average age at diagnosis of 62 years [1]. Of all women diagnosed with breast cancer 69.7% are older than 55 years [1]. The majority (81.4%) of women who die of breast cancer are aged 55 years and older with a median age of breast cancer death of 68 years [1]. Due to major improvements in public health and medical care, life expectancy has substantially increased, leading to an increase of the population age of 65 and above over the past 10 years from 37.2 million to 49.2 million and is projected to almost double to 98 million in 2060 [2]. The rising incidence of cancer with advancing age coupled with an aging population will result in an increase of cancer incidence for patients 65 years and older of 67% between 2010 and 2030 [3]. However, multiple studies have documented the underrepresentation of older adults in cancer research and trials designed specifically for older adults are rare [4,5]. Consequently, the American Society of Clinical Oncology and the Institute of Medicine have pointed out the knowledge gaps in especially older patients regarding the biological and genetic characteristics of cancer and the effectiveness and toxicities of treatment and have emphasized the need for intensified research efforts in older cancer patient groups [4].

Another tremendous clinical challenge is the considerable disparity in breast cancer patients with respect to diagnosis and prognostication, which are the basis for treatment stratification. This disparity can occur within a primary breast cancer with coexisting subpopulations of cancer cells differing in their genetic, morphological or behavioral characteristics (intra-tumor heterogeneity, ITH) and also between breast cancer in different patients (inter-tumor heterogeneity) [6]. The existence of ITH as a common phenomenon in breast cancer was documented in several studies using a variety of molecular and cytological techniques, including single cell copy number profiling by single cell sequencing or miFISH [7,8,9,10,11,12,13]. We believe that a better understanding of inter- and intratumor heterogeneity is a key challenge to advance personalized medicine towards more effective cancer treatment.

One step towards personalized medicine was achieved by adding tests based on gene expression profiling, such as OncotypeDX (Genomic Health) [14], MammaPrint [15] and EndoPredict [16], into clinical management for improved prognostication and thus more accurate treatment selection [17]. In addition to gene expression profiling the quantitative measurement of the nuclear DNA content in breast cancer cells was shown to improve prognostication as there is a clear association between the degree of aneuploidy with disease outcome [18,19]. In general, diploid tumors prove to be less malignant than their aneuploid counterparts. Furthermore, not only the status quo of nuclear DNA content, i.e., diploid or aneuploid, but also the degree of genomic instability reflected as the variability of the DNA content within the breast cancer cell population is associated with prognosis [20]. It could be shown that patients with genomically stable tumors have a significantly better prognosis than patients with genomically instable tumors [20]. However, the exact interplay between chromosomal aneuploidy, genomic instability, intra-tumor heterogeneity and disease outcome needs further elucidation.

Our study was motivated by the desire to better understand the biological and genetic features of breast cancer with an emphasis on inter- and intratumor heterogeneity, copy number alterations, genomic instability, and gene mutations especially in patients aged 50 years and older. We therefore conducted a comprehensive genetic analysis of 39 breast-cancer patients with a median age of 67 years divided into two groups with profoundly different clinical outcome (long survival, median 19 years, versus short survival, median 2.4 years). Analyzing formalin-fixed, paraffin-embedded (FFPE) archival material, we (i) investigated tumor clonality and heterogeneity by assessing CNAs of breast cancer associated genes using multiplex interphase fluorescence in situ hybridization (miFISH), a novel method developed in our laboratory that allows simultaneous enumeration of up to 20 FISH probes in individual nuclei of patient samples, (ii) determined the degree of genomic instability and tumor ploidy using miFISH and (iii) assessed the mutation status of 563 breast-cancer associated genes with targeted sequencing of a custom panel (OncoVar) [21] using Next Generation Sequencing (NGS).

## 2. Materials and Methods

### 2.1. Clinical Samples

The study was conducted according to the guidelines of the Declaration of Helsinki and approved by the local ethics committee of the University of Lübeck (#08-012 and #20-507). All breast cancer samples were collected within clinical routine diagnostics at the University Hospital Schleswig-Holstein in Lübeck, Germany, between 1989 and 1992. We selected 39 out of the collective of 245 breast cancer patients with a median age of 67 years (age range 50–85 years at the time of surgery) and with a follow-up of 22 years. We performed our comprehensive genetic analyses on formalin-fixed, paraffin-embedded (FFPE) biopsy specimens. Patients were matched for age, estrogen receptor (ER)-, progesterone receptor (PR)-, HER2 receptor and Ki67-status as well as the occurrence of metastasis but differed in overall survival. The long-survival group (*n* = 21) had a median survival of 19 years (range 13.2–21.5 years), while the short-survival group (*n =* 18) had a median survival of only 2.4 years (range 0.2–4.8 years). The staining of the ER, PR and HER2 receptor as well as the assessment of Ki67-expression levels were conducted at the Institute of Pathology at the University of Lübeck/University Medical Center Schleswig-Holstein (UKSH) using protocols established in the Institute of Pathology for analysis in clinical routine (Standard Operation Procedure VA-033, VA-050, VA-015) at the UKSH in Lübeck. Additionally, for the determination of the HER2-status, fluorescence in situ hybridization with a commercially purchased HER2/CEP17 probe was performed. Determination of ER- and PR-status as well as HER2-status was conducted according to the standards of the breast cancer guidelines of the American Society of Clinical Oncology/College of American Pathologists [22,23]. Ki-67 levels were assessed by determining the percentage of neoplastic cells that contained stained nuclei with a cut-off value of 20% [24]. The intrinsic subtypes were determined according to the ESMO Clinical Practice Guidelines published by Goldhirsch et al. [25]. The clinicopathological data of all 39 patients are summarized in Figure 1A and Appendix A.

### 2.2. Preparation of Cytospins from Archival FFPE Specimens

Cytospin slides containing single-layered interphase nuclei were prepared from macrodissected representative tumor areas of the FFPE-tissue blocks, which had been marked by a pathologist. FFPE material disintegration and cytospin preparation were done as previously published [10].

### 2.3. Multiplex Interphase Fluorescence in Situ Hybridization (miFISH)

The miFISH procedure was performed as previously described [8,10]. The following breast cancer-related eight genes were included: *COX2/PTGS2* (1q31.1), *DBC2/RHOBTB2* (8p21.3), *MYC* (8q24.21), *CCND1* (11q13.3), *CDH1* (16q22.1), *TP53* (17p13.1), *HER2/ERBB2* (17q12) and *ZNF217* (20q13.2). These genes were selected because they are frequently subject to CNAs in breast cancer, as previously determined by comparative genomic hybridization [26,27]. Two centromere probes, targeting the centromeres of chromosomes 4, (CCP4, 4p11.1-q11.1) and 10 (CCP10, 10p11.1-q11.1), were added to serve as ploidy references. Exclusively for one case (13S), additional hybridizations with further centromere (*n =* 12; CCP2, 3, 4, 6, 7, 9, 10, 11, 12, 15, 18, X) and locus-specific probes (*n =* 5; *CCNB1* (5q13.2), *RB1* (13q14.2), *CCNE1* (19q12), *DSCR8* (21q22.13), *NF2* (22q12.2)) were performed.

The individual FISH probes were custom manufactured according to our specifications by Cytotest (Rockville, MD, USA). The Fluorochrome conjugates were purchased from Dyomics (Jena, Germany) or provided by Cytotest (Rockville, MD, USA).

The FISH probes were combined into two panels containing *HER2, CDH1, TP53, ZNF217* and CCP10 (panel 1) and *COX2, CCND1, DBC2, MYC* and CEP4 (panel 2). The two panels were consecutively hybridized onto the same cytospin slide, as was a third panel with different probe colors for validation purposes for selected cases. After detection, up to 12,000 nuclei were automatically imaged with a fluorescence microscope and a 40x oil immersion objective (BX63, Olympus, Tokyo, Japan) equipped with custom optical filters (Chroma, Bellow Falls, VT, USA) using custom FISH-after-FISH software on the DUET scanning imaging workstation (BioView Ltd., Rehovot, Israel) which allows automated relocation to the same nuclei for subsequent probe panels. The custom software provides an automated image-overlay, which presents images of all 10 probes hybridized within the same nucleus in a custom gallery overview, allowing for signal enumeration of all probes per nucleus. Enumeration of all signals for a minimum of 250 nuclei for all 39 cases was performed manually in a consecutive manner and reviewed for accuracy. Only undamaged, complete nuclei, which did not overlap with another nucleus, with clearly visible signals for all 10 probes were included in the final count.

### 2.4. Determining Clonal Signal Patterns, Gain and Loss Patterns, Ploidy and Instability Index

Processing of raw data to determine clonal signal patterns, annotation of ploidy and determination of gain and loss patterns were performed as previously published [8]. After finishing the final count an excel spreadsheet containing all signal counts for each case was automatically recorded with every row of the 10 probe signals representing a signal pattern. Signal patterns which occurred in more than one counted nucleus were grouped together, with the largest group being defined as ‘the major clone’.

The cellular ploidy of each signal pattern was annotated by assessment of signal counts for CCP4 and CCP10 and a calculation of the average of all FISH probes leaving out markers with amplifications that biased the average. In the next step, an average ploidy value (decimal number) for each tumor sample was determined by calculating the average of the ploidy value of each nucleus based on the miFISH results (see Appendix A). In order to identify the most accurate cut-off value for the average ploidy for assigning each tumor sample in the group ‘diploid’ or ‘aneuploid’ based on the miFISH results we compared ploidy assignment for 34 tumor samples of an independent cohort conducted both by miFISH and DNA image cytometry (data previously published by Koçak et al. [9]). The most accurate cut-off value for determining the cellular ploidy was the following: an average ploidy value of 2.0 and 2.1 based on the miFISH results matched the ploidy-determination of a diploid case by quantitative measurement of the nuclear DNA content, an average ploidy of 2.2 and above based on the miFISH-data matched with the determination of aneuploidy by quantitative DNA measurement. We used the established cut-off value of an average ploidy of 2.2 in the miFISH results to determine diploidy and aneuploidy in the 39 cases in our study which we further confirmed by additionally performing DNA image cytometry for a subset of cases of our cohort (11 long survival- and nine short survival-cases) as described in 2.5.

Gain and loss patterns were established in relation to the assessed ploidy of the respective nucleus and visualized in a color chart for each case (Figure 2A–D, Figure 3D,H and Figure 4D,H, Appendix A). Copy number alterations (CNAs), defined as somatic changes to chromosome number that result in gains or losses in copies of DNA-sections being prevalent in many cancer types [28], were considered for statistical analyses only when the respective aberration occurred in at least 15% of the cell population (Figure 1C, Appendix A).

In order to quantify the frequency of altered clone patterns as a reflection of tumor heterogeneity, the instability index (*I*) of each case was calculated according to the following formula with *N* being the number of signal patterns observed and *n* as the number of nuclei analyzed [10].
 I=(N*100)/n

### 2.5. Quantitative Measurement of the Nuclear DNA Content by Image Cytometry

Image cytometry was performed for a subset of cases (11 long survival- and nine short survival-cases) using the ICM imaging system (Ahrens ACAS, Bargteheide/Hamburg, Germany) and Feulgen-stained cytospins as described previously [29]. All detected particles were screened in the ICM cell gallery and overlapping, or damaged nuclei were manually excluded. A minimum of 1214 nuclei for each case (mean, 6700; range, 1214 to 16,425) were analyzed. For quantitative measurement of the DNA content each of the 20 cases was screened for several diploid nuclei (granulocytes, lymphocytes) to set the 2c value indicating a diploid DNA content. The DNA values of the tumor cells were then calculated accordingly. The pattern of DNA histograms were determined according to the Auer classification [18] into diploid (Type I, III) and aneuploid (Type IV). Cell populations with an additional stem line next to the one at 2c/4c or more than 10 cells above 5c were classified as aneuploid.

### 2.6. Clonal Evolution in Tumors Assessed by Phylogenetic Tree Modelling

For each of the 39 tumors we applied phylogenetic algorithms using the software FISHtrees 3.2 in the ploidyless mode, where observed signal patterns are distinguished by their probe copy numbers [30]. FISHtrees generates a tree model starting from a normal state root (2, 2, 2, 2, 2, 2, 2) continuing by heuristically seeking to minimize the total number of CNAs across the tree. As a result, each edge moving away from the root to a new node corresponds to a change in copy number of one gene probe. The FISHtrees algorithm predicts transit signal patterns that are not observed in the sample so that in the evolution tree generated by the algorithm the up and downstream nodes can be linked. Those transit patterns are represented by nodes encircled with a dashed line. Nodes encircled by a solid line reflect miFISH signal patterns observed in the tumor sample. Additionally, we calculated the maximum tree depth, defined by the maximum number of steps away from the root node to any leaf node, and total number of events in the tree, defined by the total events needed to generate all leaf nodes from root node. Furthermore, we tested the average values for significance between the groups by two-sided t-test.

### 2.7. Targeted Next Generation Sequencing, Sequencing Data Processing, and Analyses

DNA extraction from five macrodissected 10 µm FFPE-tissue sections being lysed in a cocktail containing mineral oil for deparaffinization, proteinase K for digestion and ATL lysis buffer (Qiagen, Germantown, MD, USA) was performed as previously described [31]. 

DNA concentrations were measured on Qubit (Life Technologies, Carlsbad, CA, USA) with the dsDNA High Sensitivity Assay Kit (Life Technologies) and the DNA integrity was assessed on a Bioanalyzer (Agilent, Santa Carla, CA, USA). As an input for library preparation 200 ng DNA was used. The targeted sequence capture approach, termed OncoVar, was designed to span coding exons of 563 genes, which are related to cancer (gene list presented in Appendix A). Library construction was done with the KAPA Hyper Prep Kits for Illumina (San Diego, CA, USA) with the resulting paired-end libraries being sequenced on a NextSeq 500 system (Illumina). The mean read depth for targeted regions (mean coverage) was 263. Variant calling and data processing procedure followed the Best Practices workflow which is recommended by the Broad Institute [32].

The following filtering criteria were used for variant calling: (1) did not pass UnifiedGenotype filter with GATK default criteria, (2) fraction of alternative reads ≤5%, (3) total read depth ≤5 or alternative read depth ≤3, (4) QUAL <30, (5) low impact according to dbNSFP [33], (6) common SNPs in the NCBI dbSNP version 147 [34], (7) variants with allele frequency (AF), overall allele frequency in ESP (ESP_AF_GLOBAL), or allele frequencies (ESP6500 MAF_EA) >0.001 in the Exome Aggregation Consortium (ExAC, release 3.1) [35], (8) MAPQ score <40 for variants with ≥100 COSMIC cases or on hotspot genes of breast cancer (*TP53, PIK3CA, GATA3, MAP3K1, KMT2C*) [36], (9) MAPQ score <55 for variants on other genes, (10) variants, which exist in more than one sample including less than <100 COSMIC cases and (11) variants with no COSMIC case having ‘moderate impact’ according to dbNSFP. All identified SNVs and indels were visually validated by using the Integrative Genomics Viewer (IGV, Broad Institute [37]). Lollipop plots (Appendix A) were generated using MutationMapper [38,39].

### 2.8. Statistics

The data were statistically analyzed with regard to the differently formed subgroups by two-sided t-tests, one way Anova, Fisher exact tests and chi-square tests as appropriate to calculate the corresponding *p*-values. Multiple testing correction was done for all values using the Benjamini–Hochberg procedure. Post hoc test was done with Turkey’s HSD or Fisher exact tests with Benjamini–Hochberg corrections according to the context. After correction for multiple testing, *p* < 0.05 was considered statistically significant. For the statistical analysis of mutations analyzed by NGS, all genes showing a mutation in at least three samples (>7.5%) of the cohort were included. Additionally, a mutual exclusivity and co-occurrence analysis of mutations (*TP53, PIK3CA*) and copy number alterations (*COX2, DBC2, MYC, CCND1, CDH1, TP53, HER2, ZNF217*) was done using the Mutual Exclusivity Modules in Cancer (MEMo) algorithm from Ciriello et al. [40,41].

## 3. Results

### 3.1. Clinicopathologic Characteristics

Our cohort comprised 39 samples of patients being 50 years and above (median 67 years) and was divided into patients with long survival (median 19 years, *n =* 21) and short survival (median 2.4 years, *n =* 18). The clinical data, including survival time, age at diagnosis, tumor (T)-stage, lymph node (N)-stage, metastasis (M)-stage, ER-, PR-, Her2-status and Ki67- expression level are summarized in Figure 1A, Appendix A and shown in more detail in Appendix A. Statistical analysis did not reveal a significant difference for any of the matched parameters (age, ER-, PR-, HER2- and Ki67-status as well as the occurrence of metastasis). T1–2 stages were significantly more common in the long survival group as was the absence of lymph node metastases. Comparing the clinical parameters of diploid and aneuploid samples, no significant differences were revealed regarding age, survival time after diagnosis ER-, HER2- and Ki67-status and occurrence of lymph node and distant metastasis. However, a significant higher number of T3/4 stages and of PR-negative tumors were observed in the aneuploid group. The statistical analysis of the clinical parameters regarding samples with low versus high instability did not reveal any significant differences in any of the listed parameters (age, survival time after diagnosis, T-, N-, M-, ER-, PR-, HER2- and Ki67-status). The analysis according to intrinsic subtypes ‘luminal A/B’, ‘HER2 positive’ and ‘triple negative’ showed the longest average survival for the lumina A/B group and the shortest for the triple negative group, however, not reaching statistical significance.

### 3.2. Landscape of Gene Mutations

The somatic mutation status of 563 cancer-related genes was determined using NGS with the OncoVar panel [42]. All detected mutations are presented in Appendix A and mutated genes visualized in Figure 1B, Appendix A. Across all tumors, the most-frequently mutated genes were *PIK3CA* (12/39, 31%), which co-occurred with ER-positivity in 11 cases, and *TP53* (8/39, 21%) followed by *MAP3K1* (7/39, 18%), *KMT2C* (6/39, 15%), *CDH1* (4/39, 10%), *ITGB2* (3/39, 8%), *SPEN* (3/39), and *SF3B1* (3/39). Overall, there is an overlap between the spectrum of gene mutations observed in our cohort and the significantly mutated genes reported in The Cancer Genome Atlas (TCGA) 2012 breast cancer cohort [36], except for *ITGB2* and *SPEN*, which have not been reported as significantly mutated in the TCGA cohort and *MAP2K4*, that did not show any mutations in our cohort. We visualized the position of mutations across the protein domains for the two most-frequently mutated genes *PIK3CA* and *TP53* in our cohort in Appendix A. All mutations in *PIK3CA* occurred as missense mutations, involving the known mutation hot spots in exon 9 (E542K in cases 2L, 5L, 19L, 13S; E545K in luminal A tumors 16L, 10S), helical domain, and exon 20 (H1047R in 6L, 18L, 21L and H1047L in 5S), kinase domain with corresponding high COSMIC ID numbers [43]. The majority of mutations in *TP53* were observed as missense mutations also involving the known hot spots exclusively located in the DNA binding domain of the protein. Distributions of frequently mutated genes in different groups of the cohort, for example long versus short survival, with corresponding p-values are presented in Appendix A. We observed that *PIK3CA* was more frequently mutated in tumors with a low instability index (45%) in comparison to tumors with a high instability index (15.8%). Furthermore, more tumors of long survival patients (8/21) harbored *PIK3CA* mutations compared to tumors of patients with short survival (4/18). In addition, mutations in *SF3B1* were only revealed in long survival patients, including one missense mutation with a high COSMIC ID number. However, despite observing these trends, none of the differences in mutation frequencies of any of those genes nor the overall mutation burden per tumor between the groups of long and short survival, diploid versus aneuploid tumors or tumors with low versus high instability index reached significance. When comparing mutation frequencies for the cohort separated by the intrinsic subtypes luminal A/B, HER2 positive and triple negative a significant distribution for the gene *TP53* was revealed as shown in Appendix A (mutated in 7% of luminal A/B, in 50% of HER2 positive and in 57% of triple negative samples, *p* = 0.038).

### 3.3. Analysis of CNAs and ITH by miFISH

We have recently developed a multiplex interphase FISH technique (miFISH) providing the possibility to visualize up to 20 loci simultaneously on a single cell basis [10] and to quantitatively assess CNAs and ITH in cancer cell populations. In this study, miFISH was used to determine copy numbers of 10 loci and the ploidy of the tumor samples as described in Section 2. For a subset of cases (11 long- and nine short survival cases) we also performed DNA image cytometry. All DNA histograms are shown in the Appendix A. The ploidy results determined by DNA image cytometry and miFISH matched well, in line with our previous report (Koçak et al. 2020) [9].

Overall, all 39 cases revealed CNAs in at least two of the gene probes. The most common alterations were gains of the oncogenes *COX2* in 28/39 (72%) and *MYC* in 27/39 (69%) of the cases and losses of the tumor suppressor genes *CDH1* in 29/39 (74%) and *TP53* in 27/39 (69%) of the cases, as presented in Appendix A. An overview of all gains and losses of each gene probe and tumor sample is shown in Figure 1C. As obvious from Figure 1C, genes that were frequently gained were rarely lost, and vice versa. Consistent with previous results, the five targeted oncogenes revealed more often copy number gains while the three tumor suppressor genes were more often subject to copy number losses. Exceptions applied to the gain of *DBC2* in 9/39 cases, but, of note, in all of these cases *MYC* was gained as well. A similar picture was observed for gains of *TP53* (5/39) and losses of *HER2* (8/39), which co-occurred with the gain or loss of chromosome 17. Anti-intuitively, *CCND1* was lost in 5/39 cases. For some tumor samples, genes were not only gained but amplified, defined as exceeding more than two times the assigned overall ploidy. *HER2* presented the highest copy numbers and was amplified in seven cases (in five long and two short survival cases), followed by *CCND1*-amplification in six ER-positive cases (equally distributed in long and short survival) and *MYC*-amplification in 10 cases (equally distributed in long and short survival). In Appendix A all average signal numbers per gene probe are presented for each tumor sample.

Most cases showed both gains and losses. The miFISH results, visualized as color charts, are presented in Figure 2, Figure 3D,H and Figure 4D,H and in Appendix A. Case 13S was unusual because the tumor cell population was dominated by a clone with only one copy of *DBC2*, *MYC*, *CDH1*, *TP53* and *HER2*, respectively (Figure 2A). In addition, also the centromere probes for chromosome 4 and 10 (ploidy control probes included in the breast panels, see Section 2 showed each one copy only for most nuclei, indicating that this tumor might be hypodiploid. To confirm this hypothesis, we pursued further hybridizations with additional centromere and locus-specific probes allowing us to assess all chromosomes except for chromosome 14 as described in Section 2. The results, shown in Appendix A, revealed losses for 10 of the 22 chromosomes analyzed, further corroborating our hypothesis that case 13S is a hypodiploid tumor.

On the other hand, case 14S revealed mostly gains of *COX2*, *DBC2* and *MYC* in more than 90% of the tumor cells (Figure 2B). Case 10S was extremely instable with an instability index of 68.8 yet maintaining a gain of *CCND1* accompanied by a loss of *CDH1* in essentially all cells of the tumor (Figure 2C). The consistent gain of *CCND1* together with the loss of *DBC2* was found in three additional cases (12S, 4L, 9L). Case 9S was also remarkable: in 30% of the cells, all gene probes were subject to copy number changes, and in 58% all but one probe showed changes (Figure 2D). The case revealing the fewest CNA for the eight analyzed gene probes was 16L with a loss of the tumor suppressor genes *TP53* and *CDH1* as the sole changes (see Appendix A).

Changing the focus from the assessment of CNAs per case on a single cell basis to the overall tumor cell population based on the miFISH results, a high variability of ITH across the sample cohort was revealed. Instability indices, quantifying the frequency of altered clone patterns as a reflection of ITH (see Section 2, ranged from 2 to 86.6 (median 24.8) within the cohort (see Appendix A), giving an impression of the genetic diversity of breast cancer in women aged 50 years and older. The miFISH signals of all cases are presented in detail in Appendix A.

In Figure 3D,H two examples of long survival cases (5L and 7L) and in Figure 4D,H of short survival cases (4S and 8S) are presented as color charts. Furthermore, the histogram of the quantitative DNA content measurements and representative H&E stain of the respective case are shown in Figure 3A,B,E,F and Figure 4A,B,E,F. In the long survival case 7L 98% of the cells revealed the same gain-and-loss pattern, forming the major clone, resulting in a very low ITH (see Appendix A). In fact, case 7L has the lowest instability index of the cohort (instability index of 2) and was determined as diploid. In the pattern of the major clone in case 7L a loss of *CDH1* and *DBC2* and a gain of *MYC* was observed, which could be indicative of the formation of an isochromosome 8q [44]. In contrast, case 5L (Figure 3A–D) with an instability index of 46 and determined as aneuploid serves as an example that high ITH and aneuploidy were also observed in the long survival group. This tumor showed losses of both *TP53* and *CDH1*, along with gains of *COX2* and *ZNF217*. The short survival case 8S (Figure 4E–H) was assessed as diploid by image cytometry and with a high proliferative activity, as determined by high Ki-67 expression. In addition, case 8S revealed a stable aberration pattern dominated by consistent losses of *DBC2*, *CDH1*, *TP53* and *HER2* and gains of *MYC* and *ZNF217* in essentially all cells of the tumor, leading to a low ITH (instability index 7.6) with a major clone consisting of 95% of all cells. In contrast, case 4S (Figure 4A–D) showed a markedly increased ITH with an instability index of 54 and a major clone in 32% of the cells, containing losses of *TP53* and *ZNF217*, gains of *COX2* and *HER2* and high-level amplification of *CCND1*. The second largest clone, represented by 20% of the cells, revealed a pattern consistent with the loss of chromosome 8 and loss of *TP53* and *ZNF217.* Case 4S exhibited an aneuploid DNA histogram. In addition to the color charts, the histograms and representative H&E stains of the cases 5L, 7L, 4S and 8S we displayed the likely trajectory of clonal evolution and frequency of clones depicted in the color charts based on the miFISH results graphically as circle plots in Figure 3C,G and Figure 4C,G. The complex trajectory with several small and middle-sized circles illustrating several clones within the tumor cell population (Figure 3C and Figure 4C) represents the aneuploid cases 5L and 4S with a high instability index reflecting high ITH. In contrast, the elementary trajectory in the circle plot (Figure 3G and Figure 4G) represents the diploid cases 7L and 8S with a low instability index reflecting low ITH.

### 3.4. Genetic Characteristics of Subgroups Distinct by Survival Time, Ploidy and Instability Index

Our patients in the cohort were selected due to their profound different time of survival after diagnosis (median 2.4 versus median 19 years) aiming to gain explanations for the different prognosis based on the results using the genetic parameters. Unexpectedly, the observed CNAs and average signal numbers were not significantly different between the long and short survival groups after multiple test correction (both the overall CNA per sample and separately analyzed for the different gene probes) as presented in Figure 5C and in the Appendix A. Additionally, the instability indices as a measure of ITH were not different between the long and short survival groups with *p =* 0.7 (see Figure 5A and Appendix A).

To further analyze our miFISH results we separated the 39 patients according to their ploidy: 16 diploid cases, comprised of 10 long- and six short-survival cases and 23 aneuploid cases, comprised of 11 long- and 12 short-survival cases, as illustrated in Appendix A. We found significant differences both in the average signal numbers of each gene probe (Appendix A) and in the number of CNAs between the groups: diploid tumors had on average 4.5 CNAs, compared to 6.5 in the aneuploid tumors (*p =* 0.0012, see Figure 5D and Appendix A). Specifically, a gain of *COX2* and a loss of *TP53* were observed in 50% of diploid tumors, compared to 87% and 83% of aneuploid tumors; however, not reaching statistical significance after multiple test correction. In 22% (5/23) of aneuploid and 6% (1/16) of diploid cases *TP53* was both mutated and lost, which translates into a complete functional elimination of this tumor suppressor. Furthermore, significant differences were obvious in the instability indices (as a measure of ITH), which was higher in the aneuploid tumors (*p =* 0.0006, see Figure 5B and Appendix A). These profound differences become clearly evident when comparing the diploid cases 7L and 8S with low ITH with the aneuploid cases 5L and 4S with high ITH, visualized in Figure 3 and Figure 4 as color charts and circle plots. In addition, we separated the 39 tumors according to their instability indices into two equally sized groups and analyzed them regarding their clinical data, miFISH and NGS results (for more details see Appendix A).

The first group comprised patients with tumors exhibiting an instability index from 2 to 24.8 (average 12), the second group ranged from 25.6 to 86.8 (average 50.1). The clinical parameters were not significantly different between both groups (Appendix A). The number of CNAs was significantly higher in the group with the higher instability index, which was mainly due to copy number gains in this group (4.5 versus 6.9, *p ≤* 0.0001, see Figure 5E and Appendix A). Differences between the groups were also evident, yet not significant, with respect to mutation frequencies for *PIK3CA* with 45% in the group with low instability versus 16% in the group with high instability.

### 3.5. Phylogenetic Analysis by FISHtree Modelling

To reconstruct clonal relationships, we generated phylogenetic trees based on the miFISH data for each of the 39 cases using the software FISHtrees version 3.2. The signal pattern of each cell is shown in the phylogenetic tree starting from one diploid root cell and continuing by heuristically finding the signal pattern with the fewest CNAs. Both complex FISHtrees with multiple nodes, reflecting different signal patterns (measured by total number of distinct mutational events) in different tree levels (measured in tree depth), and simple FISHtrees with a low tree depth and a few tree edges were present in short and long survival groups. As a result, no significant differences with respect to the total number of events in the tree (*p =* 0.7, see Appendix A) and tree depth (*p =* 0.7, see Appendix A) could be seen between the two groups. In contrast, the above mentioned FISHtree parameters were profoundly different in the diploid and aneuploid tumors (total number of mutational events, *p =* 0.0008; tree depth, *p =* 0.0008; see Appendix A). As expected, the FISHtrees of the aneuploid tumors were in general more complex than the diploid ones with a multitude of nodes in different tree levels reflecting the higher burden of CNAs, increased genomic instability and higher ITH.

Similar to the comparison of diploid versus aneuploid groups, the FISHtrees parameters were also significantly different in the group of tumors with a low versus a high instability index (total number of events in the tree, *p ≤* 0.0001, and maximum tree depth, *p ≤* 0.0001, see Appendix A).

Examples of FISHtrees are presented for the long survival cases 5L and 7L in Figure 6A,B. Case 7L was diploid and consisted of two major clones according to the signal pattern, with only three additional minor nodes. Note that the FISHtree, different from the display in the color charts (Figure 3H), identified two, not one, major clones. This is attributable to the fact that the FISHtrees display the actual signal counts, whereas the color charts reflect relative gains and losses. All clones of this case showed a loss of *CDH1* and gain of *MYC*. The major clones revealed an additional loss of *DBC2*. In contrast, case 5L (Figure 6A) was aneuploid, consisted of one major clone, yet carries an enormous degree of ITH reflected in a multitude of minor clones, characterized by a high number of distinct mutational events (207) resulting in a high tree depth (15) (see Appendix A). Despite this high degree of ITH, we note that all clones had a gain of *COX2*, and more than 95% of the clones lost *CDH1* and *TP53*. As illustrated in Figure 6A, the signal patterns of individual nuclei (labeled as signal pattern 1 and 2) seen in case 5L indicate a whole genome duplication (WGD) as an initiating event resulting in a major clone with half of the markers showing signal numbers of 4, one marker showing octoploidy, and the remaining loci showing gains and losses compared to tetraploidy. In addition to the WGD as an initiating event, other WGDs presumably occurred in the four nodes 12, 15, 19 and 22, colored in pink, leading to octoploidization. We observed WGD as an initiating event in some cases of our cohort. However, more frequently a signal pattern of WGD was revealed only in a minor fraction of the tumor cell population, while the majority of clones in these cases were diploid.

The phylogenetic trees as well as the color charts and image cytometry histograms (if determined) for the remaining cases are presented in the Appendix A.

### 3.6. Mutual Exclusivity und Co-Occurrence Analyses

Mutual exclusivity analysis of copy number alterations (*COX2, DBC2, MYC, CCND1, CDH1, TP53, HER2, ZNF217*) and mutations (*TP53, PIK3CA*) was performed using the Mutual Exclusivity Modules in Cancer (MeMo) algorithm from Ciriello et al. [41]. We observed a significant co-occurrence of CNAs of *DBC2*/*MYC*, *TP53*/*DBC2* and *DBC2*/*HER2* (Appendix A). Furthermore, we detected a significant mutual exclusivity of CNAs of *HER2* and *PIK3CA* mutations and CNAs of *CCND1* and *PIK3CA* mutations.

In summary, we conducted a comprehensive analysis of 39 breast cancer patients aged 50 years and older (median 67 years) by correlating our genetic results with the survival time after diagnosis. Overall, a high variability of ITH across the cohort was revealed. Of note, the degree of ITH in patients aged 50 years and older did not correlate with prognosis. However, significant differences in CNAs and ITH could be detected when comparing diploid versus aneuploid tumors.

## 4. Discussion

Focusing on older women, as they are underrepresented in cancer research [4,5], we conducted a comprehensive genetic analysis assessing tumor ploidy and genomic instability, ITH and cancer gene mutations in breast cancers of 39 women with a minimum age of 50 years (median 67 years) and a follow-up time of 22 years. We used miFISH to enumerate copy number changes of eight breast cancer relevant genes [10] giving us the possibility to identify specific genomic imbalances for each case on a single cell level and to measure the degree of ITH by enumerating copy number differences from cell to cell which allowed us to calculate genomic instability indices.

In addition to miFISH, targeted sequence analysis was conducted to determine the mutation status of 563 cancer-related genes, using the OncoVar panel (Appendix A).

Selecting patients with long (median 19 years) and short survival (median 2.4 years) after the diagnosis of breast cancer allowed us to test the hypothesis whether the degree of ITH, genomic instability, ploidy and the landscape and frequency of mutations influences disease outcome in patients aged 50 years and older.

In our cohort of 39 patients the most common alterations were gains of the oncogenes *COX2*, *MYC*, *HER2* and losses of the tumor suppressor genes *CDH1*, *TP53* and *DBC2* identified by miFISH. These most common alterations plus the overall distribution of gains and losses of our miFISH results are consistent with previous results based on comparative genomic hybridization, i.e., the relative gains of chromosome arms or parts of chromosome arms 1q, 8q, 11q, 17q and 20q accompanied by losses of chromosome arms 8p, 16q and 17p [26,27,45] and with results reported in the TCGA database [36].

In 16/39 cases (seven long and nine short survival patients), the concomitant loss of *DBC2* and gains of *MYC*, and in 14/39 (seven in each group) the concomitant loss of *TP53* and gain of *HER2* (see Figure 1C) would be suggestive of the formation of isochromosomes 8q and 17q, respectively. Isochromosome 8q and 17q are the most common isochromosomes found in cancer [46]. When reviewing the spatial arrangement of miFISH signals for these chromosomes in interphase nuclei, we found patterns in about half of the cases that strongly suggest the presence of isochromosome formation; however, isochromosome formation could not be confirmed nor excluded in the remaining half of the cases because the signal patterns were too complex. In addition to these observations, our mutual exclusivity analyses revealed a significantly higher co-occurrence of CNAs of *DBC2* (8p) and *MYC* (8q) (q = 0.023) and also of *TP53* and *HER2* (q = 0.042) (Appendix A). In cytogenic studies, consistent 8p deletions and isochromosomes 8q were revealed as common to many human carcinomas, including breast cancer, indicating a significant pathogenetic role in tumorigenesis [44,47,48].

Some cases showed the unusual gain of a tumor suppressor gene, and/or the loss of an oncogene. To exemplify this statement the gain of the tumor suppressor gene *DBC2* was observed in 9/39 cases as visualized for cases 14S and 9S in Figure 2B,D, showing a gain of *DBC2* along with a gain of *MYC.* A possible explanation for the gain of the tumor suppressor gene *DBC2*, located on 8p, could be the gain of the entire chromosome 8 driven by the benefit of gaining a copy of the oncogene *MYC*, which resides on 8q. In fact, there was no case with a *DBC2* gain that did not show a concomitant *MYC* gain as visualized in Figure 1C, supporting this hypothesis. Similarly, prominent oncogenes, such as *HER2* (located on 17q) can be subject to loss, which in our cases was driven by the concurrent loss of the important tumor suppressor gene *TP53* (located on 17p), i.e., the loss of the entire chromosome 17. This observation is exemplified for cases 8S, 13S and 9S (Figure 4H and Figure 2A,D), which show a co-occurrence of losses of *TP53* and *HER2*. Overall, a loss of the oncogene *HER2* was revealed in eight cases, all of them showing a concomitant loss of *TP53* (see Figure 1C), supporting the hypothesis of a loss of the entire chromosome 17 as the cause for the *HER2* loss.

Overall, *HER2* presented the highest average copy number in our cohort and was amplified in seven of our 39 (18%) cases. This is consistent with a study of Riou et al., which reported *HER2* gene amplifications in 18% of breast cancers as well [49]. The *HER2* amplification in our seven cases of the cohort, in which none of the 39 patients received a targeted anti-HER2 therapy, predominantly occurred in long survival patients (five versus two). This contrasts with the literature [49,50,51,52,53]. For instance, Riou et al. reported a significant correlation of *HER2* amplification and the risk of death and the risk of multifocal distant metastases [49]. Furthermore, amplifications of the oncogene *CCND1* were revealed in six cases, equally distributed between short and long survival patients, and exclusively occurring in ER-positive breast carcinomas. This is consistent with a study of Roy et al. showing a significant correlation of *CCND1* amplification and protein expression with ER positivity [54]. In the literature, high *CCND1* amplification was associated with a poor prognosis in ER-positive cancers [54,55], which could not be confirmed in our study.

In our cohort of 39 patients, the miFISH results revealed in general an enormous degree of both inter- and intratumor heterogeneity. However, in some cases, the loss of only two tumor suppressor genes (i.e., loss of *CDH1* and *TP35* in case 16L, Appendix A), in the absence of high ITH with a low instability index of 5, was sufficient for tumor development, whereas other cases revealed copy number changes in essentially all eight genes (case 9S, Figure 2D). The degree of ITH was profoundly different among the cases. In some cases, the tumor population was dominated by a single clone (Figure 3G,H and Figure 4G,H), whereas in other cases less than 8% of the cells showed a major signal pattern (case 12L, Appendix A). Interestingly, a high number of copy number altered loci did not inevitably result in an increased number of clones nor in higher instability indices, ITH or aneuploidy. As an example, case 8S (Figure 4G,H) revealed CNAs in six of eight analyzed genes. Yet, the instability index (7.6) was low with a major clone comprising 95% of the tumor cell population indicating tumor cells with a highly aberrant, but stable karyotype.

Somewhat surprisingly, neither the distribution in diploid or aneuploid, nor the number of CNAs, nor the degree of ITH measured by the instability was notably different between the long and short survival groups, indicating that these parameters do not seem to predict survival in breast cancer patients aged 50 years and older; at least in our cohort (see Figure 5A,C). Examples for this observation are visualized in Figure 3 and Figure 4 presenting the cases 7L and 8S with a low, and the cases 5L and 4S with a high degree of instability, occurring both in the group of short and long survival, indicating that the degree of ITH in breast cancer patients aged 50 years and older does not necessarily correlate with aggressive disease. At first view, this seems to be contrary to previous studies mainly based on image cytometry that showed (for breast carcinomas of patients not selected by age) that both aneuploidy and the degree of genomic instability result in general in a poorer prognosis [18,20,56,57]. However, we should at first emphasize the fact that we exclusively studied patients with an age range of 50–85 years, hence, the higher likelihood of comorbidities could also explain these discrepancies to the results of age unbiased breast cancer patients. This contrasts to results of Cornelisse et al. [57], which includes a multivariate analysis indicating ploidy as an additional, independent prognostic factor separately evaluated for postmenopausal patients. Secondly, we realize that the eight gene probes that we used for the miFISH analysis provide only a snapshot of copy number changes in the genome. We can therefore not exclude the possibility that the inclusion of other genes would have unveiled CNAs that discern the prognostic groups, even though this is unlikely.

Additionally, we analyzed the whole cohort by dividing them (i) in the group of diploid versus aneuploid tumor samples and (ii) in the group of tumor samples with a low versus a high instability index. In the comparison between diploid versus aneuploid tumors we found significantly higher levels of CNAs in the aneuploid tumors (Figure 5D and Appendix A). Due to aneuploidy, mitosis is more complex. Chromosomal segregation errors in mitosis are the most frequent cause for chromosomal instability [58] leading to a higher number of CNAs. Furthermore, we found a significantly higher instability index as a measure of ITH in the aneuploid versus the diploid tumors being consistent with previous results [8,9] (see Figure 5B). Additionally, this correlation is also supported in the literature [58,59]. Potapova and Zhu et al. describe the relationship between aneuploidy and chromosomal instability ´as a ‘vicious cycle’, where aneuploidy potentiates chromosomal instability leading to further karyotype diversity [59]. Specifically, gains of *COX2, CCND1* and losses of *TP53* were enriched in the aneuploid tumors and so where gains of *HER2, CCND1, MYC, COX2,* and losses of *DBC2* and *TP53* in the tumors with a high instability index of our cohort (see Appendix A). Again, this is consistent with the literature [8,60,61,62,63]. However, we cannot exclude the possibility, that the higher number of T3/4-stages and of PR-negative tumors in the aneuploid group compared to the diploid group contributed to the miFISH results.

After the analysis of parameters within the different subgroups we focused on the miFISH results of notable cases (see Figure 2). Remarkably, in short survival case 13S the tumor cell population was dominated by exclusive losses of five genes, as indicated by the presence of only one instead of two signals per gene-probe in the miFISH analysis (Figure 2A). Furthermore, losses for 10 of 22 chromosomes analyzed were observed leading to the postulation of a severely hypodiploid tumor (Appendix A). Hypodiploidy in breast cancer has been reported in 1–2% of breast cancers, associated with poor prognosis [64]. The patient 13S deceased 12 months after diagnosis. Tanner et al. showed a significant correlation of a *CCND1*-gain and hypodiploidy in breast carcinomas [64]. However, a gain of *CCND1* was not revealed in our hypodiploid case 13S. Moreover, the case 10S (Figure 2C) was peculiar as well. Here, a loss of *CDH1* along with a gain of *CCND1* was present in 99% of cells, despite an enormous degree of seemingly random chromosomal instability leading to several minor clones and a high instability index (68.8). We interpret these findings such that this tumor is exclusively driven by the loss of *CDH1* and gain of *CCND1*. Otherwise, one would have expected that other, for tumorigenesis beneficial gains and losses, for instance the gain of *COX2* or the loss of *TP53*, which are present in minor clones of 10S, would have become major clones under the selective pressure. This interpretation is supported by literature as cancer genomes are described as principally dynamic and changing under selective pressures during tumorigenesis [58,65,66]. We conclude that breast cancer development does not require copy number changes of several breast cancer genes but that the deregulation of specific pathways can be sufficient.

To understand more about tumor evolution mechanisms, we generated FISHtrees based on computational reconstruction of tumorigenesis using our miFISH results. The enormous degree of ITH was reflected in the corresponding FISHtrees as some revealed simple patterns with only a few branches and nodes (exemplified for the diploid, highly stable case 7L, Figure 6B) and others revealing a high number of distinct mutational events and tree depth (visualized for aneuploid case 5L with a high instability index, Figure 6A). Interestingly, Case 5L also showed signal patterns consistent with a whole genome duplication and mitotic catastrophe, also referred to as ‘big bang’ as the initiating event [67,68]. In a study of Bielski et al. WGD is concluded to be highly common in cancer and ‘a macro-evolutionary event associated with poor prognosis´ [69]. Next to the WGD as an initiating event in case 5L, WGDs presumably occurred additionally in the four nodes 12, 15, 19 and 22 (colored in pink). As these four nodes are located downstream of node 1 in the FISHtree, these WGDs presumably occurred later in tumor progression according to the FISHtree algorithm. Interestingly, these WGDs did not lead to an expansion of these clones, indicating, that WGD does not necessarily render a selective advantage. This is supported by a previous study of younger breast cancer patients in which WGD was shown to occur but was not required for tumorigenesis [9]. Furthermore, it is consistent with a study of Lei et al. describing WGD as neither necessary for tumorigenesis nor necessarily a one-time event in cancer evolution [70].

Overall, we observed mutation spectra similar to the most-frequently reported mutations in the TCGA database [36] leading to the conclusion that the overall mutation spectrum resembles the one from an age unbiased cohort of patients with breast cancer. When comparing the overall mutation burden per tumor sample or mutation frequencies analyzed for each of the eight analyzed genes between the groups of the cohort (long versus short survival, diploid versus aneuploid tumor samples, samples with low versus high instability index) no significant differences could be detected.

Consistent with the literature [36], *PIK3CA,* encoding for the catalytic subunit p110α of class IA PI3-kinas, was the most-frequently mutated gene in our breast cancer study. The mutations in *PIK3CA* occurred predominantly (10 of 13 cases) in the three mutation hot spots of exon 9 and 20 [43] with the hot spot mutation E545K being observed exclusively in luminal A tumors (case 16L and 10S) as described in the literature [36]. For these mutations at hot spot amino acid residues oncogenic effects have been shown [71,72]. Furthermore, consistent with literature, the *PIK3CA* mutations observed in our cohort were more common in the long survival group and co-occurred with ER-positivity [36,73]. In line with the Catalog of Somatic Mutations in Cancer (COSMIC) database [74] in 21% (eight of 39) cases of our breast cancer cohort a mutation in the tumor suppressor gene *TP53* was revealed (COSMIC: 23%), including some of the hot spot mutation sites. Furthermore, the distribution of *TP53* mutations was significantly linked to the intrinsic subtypes with the highest mutation rate in triple negative samples and the lowest in luminal A/B tumor samples, consistent with literature [75,76].

## 5. Conclusions

In this study, we analyzed tumor heterogeneity, ploidy and genomic instability in breast cancer patients aged 50 years and older by taking into account their survival time after diagnosis. In conclusion, the pattern of CNAs of the eight analyzed cancer-related genes and the distribution of gene mutations of the OncoVar panel observed in breast carcinomas of patients aged 50 years and older are similar to an age-unbiased cohort. Of note, neither the number of CNAs nor tumor ploidy nor the degree of ITH showed a correlation with disease prognosis (distribution to the groups with short or long survival) for breast cancer patients aged 50 years and older. We interpret these findings such that in our group of breast cancer patients with a median age of 67 years these parameters do not seem to have an effect on survival, perhaps due to the fact that comorbidities and age at diagnosis play a major role with regard to individual prognostication.

Overall, we detected a large variability of genomic instability profiles and inter- and intratumor heterogeneity, yet maintenance of breast cancer specific chromosomal gains and losses and of frequently observed cancer gene mutations. Significant differences were found by comparing the miFISH results of diploid versus aneuploid tumor samples: aneuploid tumors showed significantly higher average signal numbers, CNAs and instability indices.

## Figures and Tables

**Figure 1 cancers-13-03366-f001:**
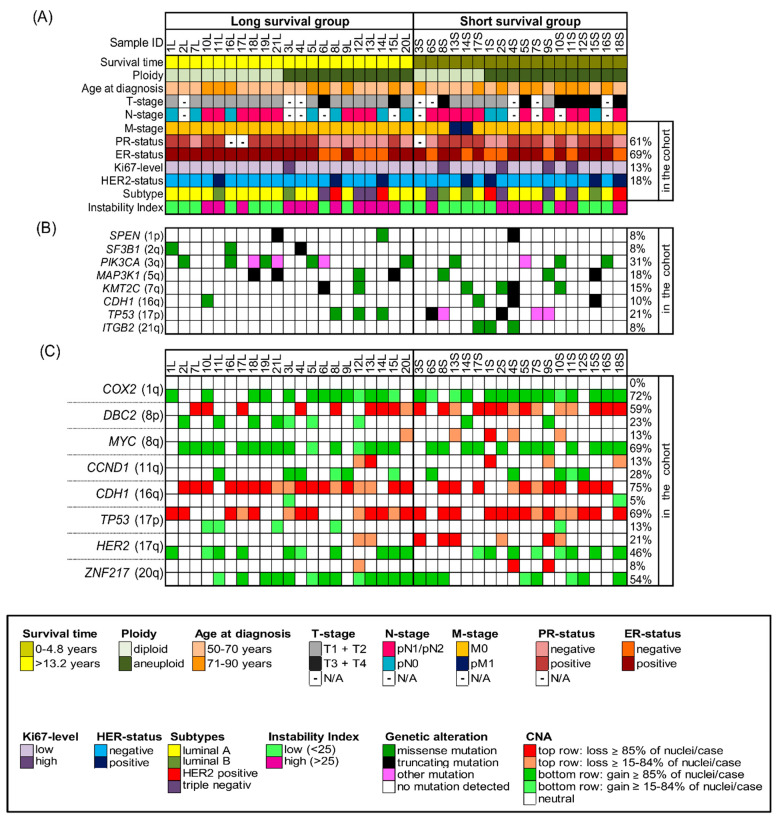
Clinicopathological features, NGS-mutation-analysis and miFISH results presented for the entire cohort (*n =* 39) with corresponding color codes sorted by ploidy and separated into the groups of long survival and short survival after diagnosis. (**A**) Summary of the clinicopathological features (rows) of the short and long survival groups, additionally sorted by ploidy and plotted per individual sample (columns). (**B**) Extraction of mutation analysis of 563 breast-cancer associated genes (OncoVar) by NGS. Distribution of mutations sorted by chromosomal location that affected genes in at least three samples. The color code indicates the type of mutation. *PIK3CA*, *TP53* and *MAP3K1* were the most-frequently mutated genes. (**C**) Copy number alterations of 8 breast cancer-related genes sorted by chromosomal location were identified by miFISH and are plotted vertically per individual sample (columns). Green color indicates a gain, red color a loss. We chose dark hues if the majority (≥85%) of all nuclei showed a gain/loss and light hues if 15–84% showed a gain/loss. The oncogenes *COX2* and *MYC* and the tumor suppressor genes *CDH1* and *TP53* were most-frequently subject to copy number alterations.

**Figure 2 cancers-13-03366-f002:**
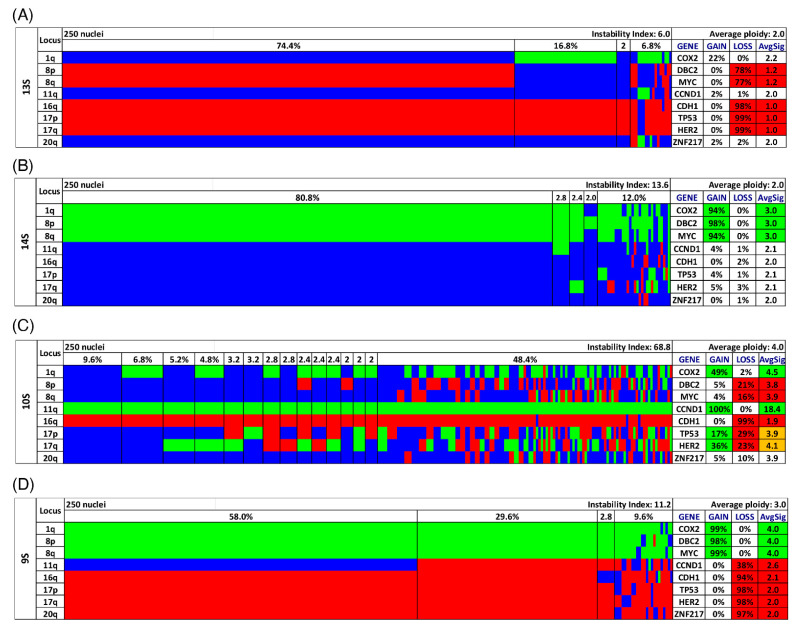
Color charts of miFISH analysis with 8 gene-specific probes of 4 notable short survival cases. Copy number counts for each nucleus are displayed as gains (green), losses (red) and neutral (blue). Gene-specific miFISH markers are plotted vertically with the ‘Locus’ column depicting the specific chromosome arm for each probe on the left of the plot, and the corresponding gene name on the right. Nuclei are arranged horizontally by the frequency of signal patterns from left to right. Each vertical line discerns specific gain-and-loss patterns and the prevalence of the cell clone in the tumor population. Copy number gains and losses are depicted as percentages of the total cell population in the ‘Gain’ and ‘Loss’ column of the table on the right. Color-labeled percentages indicate a threshold of 15%. The column ‘AvgSig’ refers to the average of all signal numbers specified for each of the 8 analyzed gene probes. Orange labeled AvgSig-values indicate that the threshold value of 15% of all nuclei was reached for both a detected copy number gain and loss in the respective gene probe. (**A**) Case 13S. The case 13S is dominated by several losses of the 8 gene probes for most nuclei, (**B**) Case 14S. The case 14S reveals mostly gains for the majority of nuclei, also a gain of the tumor suppressor gene *DBC2*, (**C**) Case 10S. The case 10S is extremely instable yet maintains a gain of *CCND1* accompanied by a loss of *CDH1* in essentially all nuclei. (**D**) Case 9S. The case 9S shows in 30% of the analyzed nuclei in all 8 gene probes copy number changes and in 58% of the nuclei in 7 of the gene probes.

**Figure 3 cancers-13-03366-f003:**
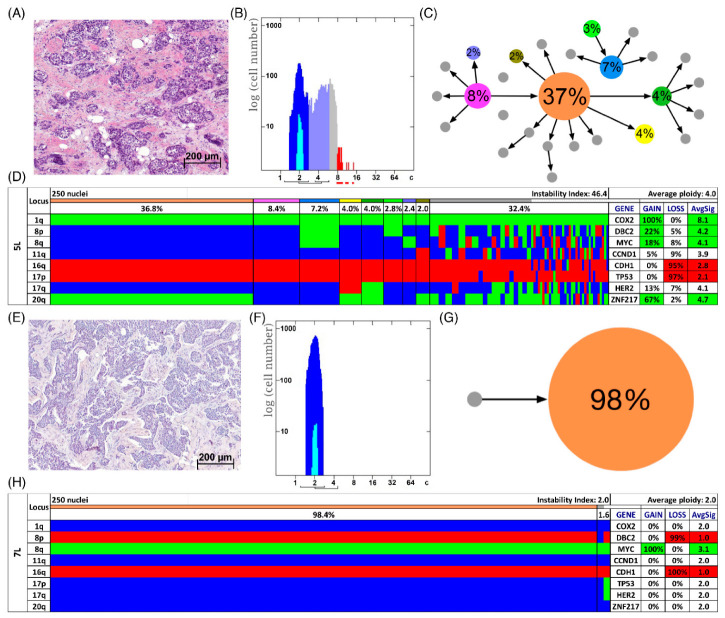
Patients with long survival. Histology (**A**,**E**), image cytometry (**B**,**F**), imbalance clone plots (**C**,**G**) and miFISH results (**D**,**H**) for the cases 5L (**A**–**D**) and 7L (**E**–**H**). (**A**,**E**) Sections of the respective breast cancer samples showing the histomorphology based on H&E staining. (**B**,**F**) DNA histograms showing the quantitative measurements of the nuclear DNA content assessed by image cytometry using Feulgen-stained cytospins. The DNA measurements revealed aneuploidy in case 5L and diploidy in case 7L. For quantitative measurement of the DNA content the sample was screened for several diploid nuclei (granulocytes, lymphocytes) to set the 2c value indicating a diploid DNA content. The quantitative measurements of the nuclear DNA content (x axis) of the tumor cells given in ‘c’ units were then calculated accordingly [18]. The y axis represents the total cell count. In case 5L 4759 and in case 7L 11,792 nuclei were analyzed. (**C**,**G**) Imbalance clone plots visualizing the clonal composition of the analyzed breast cancer section and their putative evolutionary trajectory. The area of the circles correlates with the occurrence of a cell-clone with a specific gain-and-loss pattern within the tumor cell population. Clones derived by a single gain or loss change are connected by arrows. The arrows indicate the clonal evolution according to gain-and-loss patterns in the color charts (**D**,**H**) starting from the clone with the fewest gains and losses. Thus, unconnected clones must differ in more than one gain or loss in their gain-and-loss pattern. Color coding allows assignment of the individual clones to the corresponding clones in the color charts in D and H. (**D**,**H**) Color chart of miFISH analysis with 8 gene-specific probes. Copy number counts for each nucleus are displayed as gains (green), losses (red) and neutral (blue). Markers are plotted vertically with the ‘Locus’ column depicting the specific chromosome arm for each probe on the left of the plot, and the corresponding gene name on the right. Nuclei are plotted horizontally by pattern frequency. Each vertical line discerns specific gain-and-loss patterns and the prevalence of the cell clone in the tumor population. Copy number gains and losses are depicted as percentages of the total cell population in the ‘Gain’ and ‘Loss’ column of the table on the right. Color-labeled percentages indicate a threshold of 15%. The column ‘AvgSig’ refers to the average of all signal numbers specified for each of the 8 analyzed gene probes.

**Figure 4 cancers-13-03366-f004:**
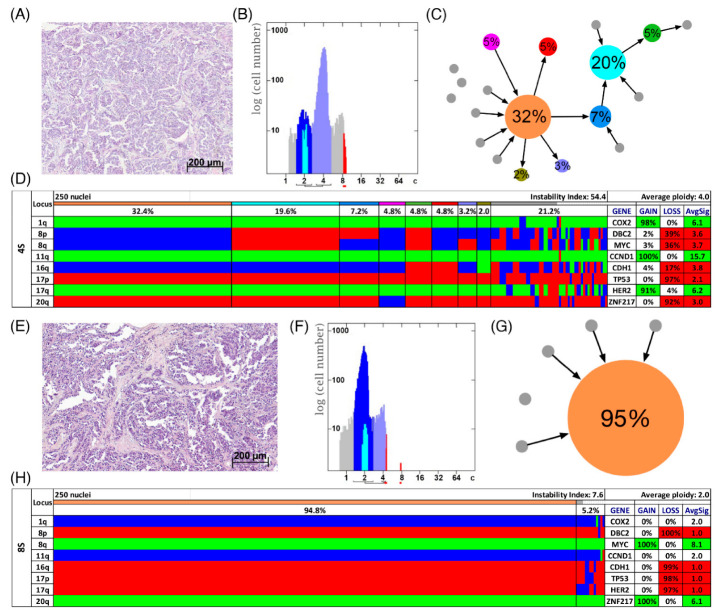
Patients with short survival. Histology (**A**,**E**), image cytometry (**B**,**F**), imbalance clone plots (**C**,**G**) and miFISH results (**D**,**H**) for the cases 4S (**A**–**D**) and 8S (**E**–**H**). (**A**,**E**) Sections of the respective breast cancer samples showing the histomorphology based on H&E staining. (**B**,**F**) DNA histograms showing the quantitative measurements of the nuclear DNA content assessed by image cytometry using Feulgen-stained cytospins. For quantitative measurement of the DNA content the sample was screened for several diploid nuclei (granulocytes, lymphocytes) to set the 2c value indicating a diploid DNA content. The quantitative measurements of the nuclear DNA content (x axis) of the tumor cells given in ‘c’ units were then calculated accordingly [18]. The y axis represents the total cell count. In case 4S 5810 and in case 8S 6721 nuclei were analyzed. (**C**,**G**) Imbalance clone plots visualizing the clonal composition of the analyzed breast cancer section and their putative evolutionary trajectory. The area of the circles correlates with the frequency of a cell-clone with a specific gain-and-loss pattern within the tumor cell population. Clones derived by a single gain or loss change are connected by arrows. The arrows indicate the clonal evolution according to gain-and-loss patterns in the color charts (**D**,**H**) starting from the clone with the fewest gains and losses. Thus, unconnected clones must differ in more than one gain or loss in their gain-and-loss pattern. Color coding allows assignment of the individual clones to the corresponding clones in the color charts in D and H. (**D**,**H**) Color chart of miFISH analysis with 8 gene-specific probes. Copy number counts for each nucleus are displayed as gains (green), losses (red) and neutral (blue). Markers are plotted vertically with the ‘Locus’ column depicting the specific chromosome arm for each probe on the left of the plot, and the corresponding gene name on the right. Nuclei are plotted horizontally by pattern frequency. Each vertical line discerns specific gain-and-loss patterns and the prevalence of the cell clone in the tumor population. Copy number gains and losses are depicted as percentages of the total cell population in the ‘Gain’ and ‘Loss’ column of the table on the right. Color-labeled percentages indicate a threshold of 15%. The column ‘AvgSig’ refers to the average of all signal numbers specified for each of the 8 analyzed gene probes.

**Figure 5 cancers-13-03366-f005:**
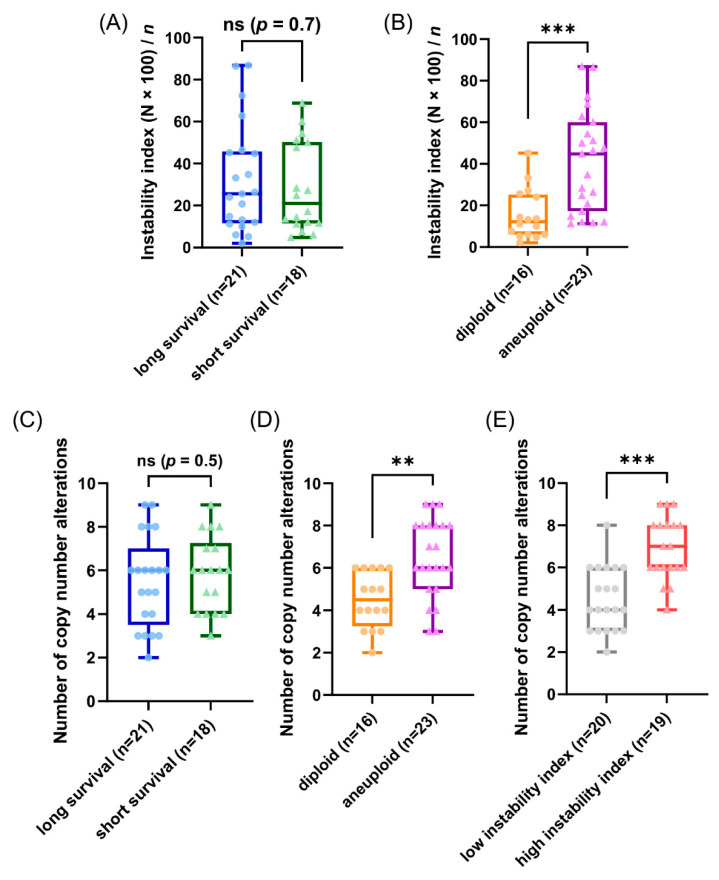
Results of the miFISH analysis presenting instability indices, calculated as described in Section 2, and frequencies of CNAs per tumor sample for different subgroups. (**A**,**B**) Instability indices including minimum, maximum, median and outliners are presented as a boxplot for each subgroup: (**A**) long and short survival and (**B**) diploid versus aneuploid tumors. Note the significant difference between diploid versus aneuploid tumors (*p* = 0.0006). (**C**–**E**) Frequency of CNAs per tumor sample including minimum, maximum, median and outliners are presented as a boxplot for each subgroup: (**C**) long and short survival, (**D**) diploid versus aneuploid tumors and (**E**) tumors with low versus high instability index. Note the significant difference between diploid versus aneuploid tumors (*p* = 0.0012) and between tumors with low versus high instability index (*p* ≤ 0.0001); ** *p* ≤ 0.01; *** *p* ≤ 0.001.

**Figure 6 cancers-13-03366-f006:**
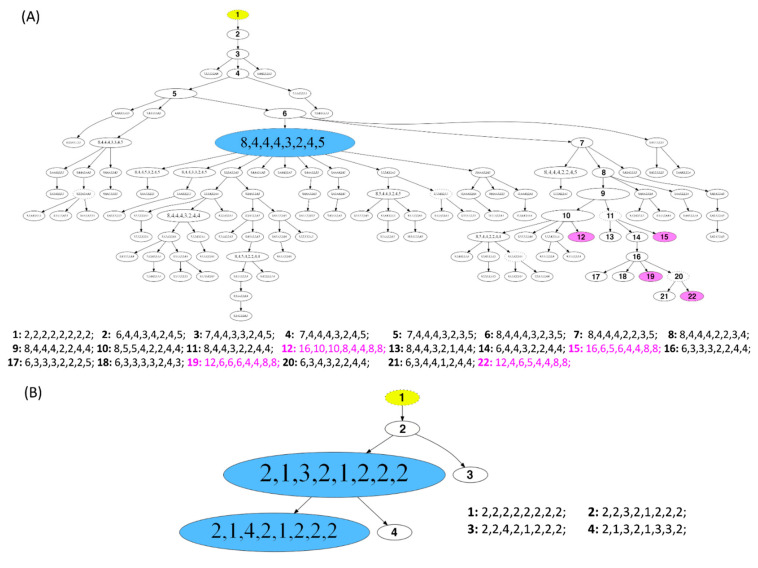
Phylogenetic trees of long survival cases (**A**) 5L and (**B**) 7L. The construction of the FISHtrees was done using phylogenetic algorithms (software FISHtrees 3.2) in the ploidyless mode. FISH patterns are depicted in the following gene order *COX2*, *DBC2*, *MYC*, *CCND1*, *CDH1*, *TP53*, *HER2*, *ZNF217*. The FISHtrees results show the clonal evolution by generating a tree model starting from a normal state root labeled in yellow color (2-2-2-2-2-2-2-2) continuing by heuristically seeking to minimize the total number of CNAs across the tree. The size of the nodes reflects, but is not proportional to, the frequency of the patterns in the cell population. Nodes encircled by a solid line reflects miFISH-signal-patterns observed in the tumor sample. The FISHtrees algorithm predicts transit signal-patterns that are not observed in the sample so that in the evolution tree generated by the algorithm the up and downstream nodes can be linked. Those transit patterns are represented by nodes encircled with a dashed line. Details of the analysis are described in Section 2. (**A**) The blue labeled node indicates the presence of the tetraploid major clone (pattern 8-4-4-4-3-2-4-5; consisting of 71 out of 250 analyzed nuclei). Detailed signal patterns for selected clones (1–22) are displayed in the legend. In 4 clones, whole genome duplication events occur, marked in pink color with the numbering 12, 15, 19 and 22 (patterns: 16-10-10-8-4-4-8-8; 16-6-5-6-4-4-8-8; 12-6-6-6-4-4-8-8; 12-4-6-5-4-4-8-8). (**B**) The blue labeled nodes indicate the presence of the major clone with the most common pattern (2-1-3-2-1-2-2-2; consisting of 217 out of 250 analyzed nuclei) and the clone with the second most common pattern (2-1-4-2-1-2-2-2; consisting of 29 out of 250 analyzed nuclei). Detailed signal patterns for selected clones (1–4) are displayed in the legend.

## Data Availability

The data sets used and/or analyzed during the current study are available from the corresponding author on reasonable request.

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
