# Peer review of "Single Cell Genetic Profiling of Tumors of Breast Cancer Patients Aged 50 Years and Older Reveals Enormous Intratumor Heterogeneity Independent of Individual Prognosis"

_cancers, 2021, doi:10.3390/cancers13133366_

Round 1
Reviewer 1 Report
The study from Liegmann and Ried et al, “Single cell genetic profiling of tumors of older breast cancer patients reveals enormous intratumor heterogeneity independent of individual prognosis,” is a well-constructed and written study. The authors attempt to uncover a link between survival times of breast cancer patients and the genetic profiles of their tumours through single cell technologies. Although the study falls short of finding significant causes for different survival times, they have not overstated results, not hidden any findings and reported accurately exactly what they have found. This kind of study is difficult to write up and I commend the authors for their work and contribution given the results. In addition, their intratumor heterogeneity findings are interesting and may allow this factor to be ruled out of breast cancer prognosis. I would suggest some edits to the manuscript before publication.
Additional Experiments:
- I would suggest that the authors stratify their patients based on receptor expression status. Given that there are only 39 patients, try TNBC (ER/PR/HER -), HER2 (HER+) and Luminal (ER/PR+). Subtypes are very important in breast cancer and due diligence would be to test if any associations exist between subtypes and survival, as TNBC/Basal-like tumours tend to have poorer outcomes without targeted therapies. I understand this may not yield any results, but is important to cover and include in the manuscript.
Minor Comments:
- There are a lot of data and supp material which is great and appreciated, but if the authors could add to figure 1, on the right-hand side the percentages of various markers and mutations in the cohort (even though it is in the supp data), that might make it a bit easier to read.
- In Figure 5, change the colours to be different for different stratifications, i.e. blue and green for survival and two other colours for diploid and aneuploid, just makes that figure more clear.
- Supp Figures 3&4 and 1&2 are out of order
- Figure 3B, F, 4B, F, y-axis labels
- Overall figures, if the authors could slightly increase the font sizes that would help, especially for the colours charts of miFISH.
Reviewer 2 Report
Summary:
This work describes the significant inter- and intratumor heterogeneity within older breast cancer patients. Investigations including older breast cancer patients are important as this patient population represents the majority of women diagnosed with and dying from breast cancer. Moreover, the null findings that neither copy number alterations, tumor ploidy, specific gene mutations, nor genomic instability correlate with prognosis are important to report as they can help prioritize those clinicopathologic characteristics that do correlate with prognosis, such as tumor staging.
Minor concerns:
- The imbalance clone plots in figures 3C,G and 4C,G are redundant based on the data presented in figures 3D,H and 4D,H. Likewise, given that the phylogenetic trees presented in figures 6A,B and the supplemental data display more detailed illustrations of the phylogeny of these clonal imbalances, the inclusion of the imbalance clone plots is repetitive and unnecessary.
- The discussion of single gene mutations that occurred in only one (or 3) cases and their connection to the literature in lines 827-845 is unjustified (particularly with n=1) and unnecessary given that the goal of the study was to provide a comprehensive picture of genomic instability.
- Figures 5B,D,E illustrating that a high number of CNAs corresponds with high genomic instability and aneuploidy corresponds with high instability and high CNAs is just intuitive and does not represent meaningful findings for the current study. I suspect these “significant” findings would hold true in the genomic profiling of any tumor subsets.
- Indeed, beyond showing that genomic instability and number of CNAs do not correlate with survival time in older breast cancer patients, none of the data presented or described in section 3.4 significantly influences the interpretation of the results. Thus, the two full paragraphs describing the non-significant findings presented in the supplementary data could be removed.
- The single sentence in lines 738-741 doesn’t function well as a stand-alone paragraph.
- Line 796 refers to case 5L being illustrated in figure 6B but it is in 6A.
- Line 850 should say eight.
Reviewer 3 Report
The authors examined genomic instability profiles of older breast cancers by miFISH and NGS.
Although the authors gave a lot of data, the manuscript is not suitable for publication in Cancers.
First of all, the authors did not seem to focus on older breast cancers. Although the samples were collected from the odder patients, comparison with young patients was not performed. Furthermore, the definition of "older" is not given.
Next, the manuscript is too descriptive and the novelty is not clearly indicated. In addition, the authors did not demonstrate significant difference between shorter and longer survival patients.
Round 2
Reviewer 3 Report
Please define "older" by providing appropriate references.
For example, in the previous study, "older" is defined as ≥70 years (https://doi.org/10.1111/ecc.13357) or ≥65 (https://doi.org/10.1016/j.ijrobp.2017.02.001).
In the present manuscript, Median age of older is67 years (age range 50-85 years), while median age of younger patients is 40 years (range, 31 to 59), according to your previous study by Koçak et al. Range 50-59 is overlapping and it is very confusing. It is scientifically not acceptable to use different definition among the manuscript.
Author Response
Please see attach file
